# Comparative Analysis of Original and Replaced Gut Microbiomes within Same Individuals Identified the Intestinal Microbes Associated with Weight Gaining

**DOI:** 10.3390/microorganisms10051062

**Published:** 2022-05-20

**Authors:** Chongkai Zhai, Ji-Seon Ahn, Md Minarul Islam, Enkhchimeg Lkhagva, Hea-Jong Chung, Seong-Tshool Hong

**Affiliations:** 1Department of Biomedical Sciences and Institute for Medical Science, Jeonbuk National University Medical School, Jeonju 54907, Korea; zhaichongkai@gmail.com (C.Z.); ajs0105@kbsi.re.kr (J.-S.A.); mislambcmb@gmail.com (M.M.I.); enkhchmg2580@gmail.com (E.L.); 2Animal Diseases and Public Health Engineering Research Center of Henan Province, Luoyang Polytechnic, Luoyang 471023, China; 3Gwangju Center, Korea Basic Science Institute, Gwangju 61751, Korea

**Keywords:** body weight gain, obesity, gut microbiome, Firmicutes, Bacteroidetes, comparative analysis, gut microbiome replacement, same individual, comparison, *Trichinella pseudospiralis*

## Abstract

The precise mechanisms of action of the host’s gut microbiome at the level of its constituting bacteria are obscure in most cases despite its definitive role. To study the precise role of the gut microbiome on the phenotypes of a host by excluding host factors, we analyzed two different gut microbiomes within the same individual mouse after replacing the gut microbiome with a new one to exclude the host factors. The gut microbiome of conventional C57BL/6 mice was randomly reestablished by feeding fecal samples from obese humans to the mice, and depleting their original gut microbiome with an antibiotic and antifungal treatment. Comparison of body weight changes before and 3 months after the replacement of the gut microbiome showed that the gut microbiome replacement affected the body weight gain in three different ways: positive, medium, and negative. The differences in body weight gain were associated with establishment of a different kind of gut microbiome in each of the mice. In addition, body weight gaining was negatively associated with the Firmicutes/Bacteroidetes ratio, which is consistent with previous recent findings. Thorough statistical analysis at low taxonomic levels showed that uncultured bacteria NR_074436.1, NR_144750.1, and NR_0421101.1 were positively associated with body weight gain, while *Trichinella pseudospiralis* and uncultured bacteria NR_024815.1 and NR_144616.1 were negatively associated. This work shows that replacement of the gut microbiome within the same individual provides an excellent opportunity for the purpose of gut microbiome analysis by excluding the host factors.

## 1. Introduction

A typical human carries a complex microbial community containing 100 trillion microbes in the digestive tract as a gut microbiome. Because the gut microbiome affects almost all human phenotypes including diseases, the gut microbiome and its role in the phenotypes and diseases of humans has become a rapidly growing area of research [1]. Failure in functional interactions between the human body and the gut microbiome has been well-proven to cause gastrointestinal (GI) tract disorders such as inflammatory bowel disease, cholelithiasis, constipation, ulcerative colitis, Crohn’s disease, etc. [2,3]. In addition to GI tract diseases, the gut microbiome plays significant determinant roles in autism, obesity, diabetes, atherosclerosis, metabolic syndrome, various neurodegenerative diseases such Alzheimer’s disease, depression, intelligence, mood, etc. [1,4,5,6].

Despite the determinant roles of the gut microbiome in human phenotypes, elucidation of the mechanistic pathway has not moved forward as expected because of limitations in identifying the causative intestinal microbes responsible for the phenotypes. Obesity is a typical example. Previous research on the relationship between obesity and the gut microbiome was focused on large taxonomic levels (phyla, classes, orders) because low taxonomic levels did not show statistical significance. Original research showed that obesity was determined by the Firmicutes/Bacteroidetes ratio and that the high prevalence of Firmicutes was associated with obesity [7]. The prevalent association of Firmicutes with obesity has been supported by subsequent research [7,8,9]. However, contrary to these results, a number of studies showed an opposite ratio of the two groups of bacteria with obesity or lack of association with any of these bacteria [10,11,12,13,14]. The discrepancy at large taxonomic levels could stem from the fact that bacterial species within a large taxonomic level are heterogenous in terms of their functions. It would obviously be unreasonable to expect that the bacteria within a large taxonomic level behave similarly. In addition, the host factors make it difficult to find the individual intestinal microbes responsible for affecting obesity. Considering the functional heterogeneity of individual bacteria within the same large taxonomic level, the discrepancy in the Firmicutes/Bacteroidetes would be natural. In this context, investigation of the gut microbiome at low taxonomic levels would hold the key to the secret about the determinant roles of the gut microbiome in respect of human phenotypes.

Since most mammalian phenotypes are determined by the interaction between host factors and the gut microbiome [3], the host factors would contribute as disturbing factors in analyzing the effects of the gut microbiome in the phenotypes. Even in gnotobiotic animal experiments, the effect of the host factors cannot be excluded because the genetic backgrounds of individual germ-free animals differ from each other. To analyze accurately the effect of the gut microbiome on the phenotype of a host by excluding host factors, the different sets of gut microbiomes should be compared within the same individual. If the two different sets of gut microbiomes are compared within the same individual, the host factors can be excluded during the analysis because the gut microbiomes are present within the genetically identical host. Because comparative analysis of before and after gut microbiome modification within the same individual excludes host factors that are critical obstacles, comparison of the original microbiome with the replaced microbiome would pinpoint intestinal microbes playing determinant roles in the phenotypes of hosts.

Despite the necessity, exclusion of genetic factors in gut microbiome analysis has not been tried. In this study, we developed a method which excluded the host factors during the analysis of the gut microbiome. To exclude the host factors, the original gut microbiomes of normal conventional mice were disturbed by antibiotic treatment following feeding with fecal samples from human obese individuals in an attempt to form a new gut microbiome. The comparative analysis of the original gut microbiome with the newly established gut microbiome together with body weight changes successfully identified the intestinal microbes associated with weight gaining at the species level.

## 2. Materials and Methods

### 2.1. Study Design and Animal Experiments

Twenty C57BL/6 mice at 12~18 months old were purchased from Korea Basic Science Institute (Gwangju, Korea). The mice were individually housed in an SPF facility with access to sterilized chow diet and water. They were maintained on a 12 h light–dark cycle at a temperature of 22 ± 1 °C and humidity of 40~50%. One week after adaptation, the mice were weighed, and the feces and blood of the mice were individually collected. After that, all mice were allowed to drink water containing three antibiotics (1 g/L ampicillin, 0.5 g/L kanamycin, and 0.5 g/L cefoxitin; Sigma-Aldrich, St. Louis, MO, USA) and an antifungal (0.5 g/L nystatin) for one week to deplete their endogenous microbiota.

About 1 g of fecal samples from 10 overweight volunteers with a BMI index of 25.6~29.1 who visited the lab were collected in 10 mL JMS medium [15]. The volunteers were 6 females and 4 males, and the ages were in the range 23~45 years old. The fecal samples were maintained at 4 °C for 0~3 days before mixing them. After mixing the fecal samples, 20 μL of the fecal medium was administered by the oral gavage method twice a week for 2 weeks in order to transplant human intestinal microbes into the mice. All of the mice were weighed after finishing feeding of the human fecal sample, and the feces and blood of the mice were individually collected.

After finishing the replacement of the native gut microbiome with the human microbiome in the mice, the mice were maintained in the above normal mouse facility for 10 weeks. The mice were weighed again on the final day of the experiment, and the feces and blood of the mice were individually collected. The mice were able to be grouped into three groups after the 3 month experimental period based on body weight changes: the positive group, which gained 5.94 ± 1.11 g in body weight after replacing with the human gut microbiome (positive body weight (PBW), *n* = 5); the medium group, which gained 2.55 ± 0.13 g in body weight (medium body weight (MBW), *n* = 6); and the negative group, which lost 1.62 ± 0.77 g in body weight after replacing with the human gut microbiome (negative body weight (NBW), *n* = 6).

### 2.2. Biochemical Analysis

The serum levels of total cholesterol (TCHO), triglyceride (TG), and high-density lipoprotein cholesterol (HDL-CHO) were determined by enzymatic methods using commercial assay kits (Asan Pharmaceutical Co., Seoul, Korea) as described previously [16]. In brief, the low-density lipoprotein cholesterol (LDL-CHO) levels were calculated using Friedewald’s equation [(LDL-CHO) = (TCHO) − ((HDL-CHO) − (TG)/5)].

Blood glucose was measured using a portable blood glucose meter (Accu-Chek Active; Roche Diagnostic GmbH, Mannheim, Germany).

### 2.3. Micro-Computed Tomography (Micro-CT)

The fat volume was determined by micro-CT as described previously [17]. In brief, micro-CT scanning was performed using a high-resolution in vivo micro-CT system (Skyscan, Konitch, Belgium) at the Center for University-wide Research Facilities (CURF) of Jeonbuk National University to analyze the mice’s fat volume. The abdominal fat regions from lumbar vertebrae 1 to 5 were analyzed, micro-CT images were taken, and total fat volume was analyzed using the Skyscan software (Skyscan 1076, Bruker. microCT, Kontich, Belgium), CT Analyzer.

### 2.4. Histological Analysis

The histological analysis was determined by a method as described previously [18]. Briefly, liver and adipocyte tissue samples were collected from mice at month 3 of the experimental intervention and were used for microscopic analysis after being stained with H&E. Immediately after isolation, liver and adipocyte tissue sections were fixed at 10% neutral buffered formalin and embedded in paraffin. Tissues were sectioned at 6 µm thickness. Paraffin in the tissue sections was removed with hot water. The tissue sections were placed on microscopic slides, and the slides were air-dried and stored overnight at 65 °C. Finally, the tissue sections were stained with hematoxylin and eosin (Vector Laboratories Inc., Burlingame, CA, USA) according to standard laboratory procedure. The H&E-stained tissue sections were observed under a light microscope (AmScope, T690C-PL), and images were taken with a microscopic digital camera (AmScope, MU-1803).

### 2.5. 16S rRNA Gene Sequencing

Total bacterial genomic DNA from each sample, before and after replacement of the gut microbiome, was extracted using the phenol:chloroform:isoamylalcohol extraction method as described previously [4,19]. Briefly, the bacterial samples suspended in lysis buffer were broken using a bead beating homogenizer. Genomic DNA was isolated from the aqueous phase during the phenol-chloroform extraction procedure. The genomic DNA was precipitated by adding 3M sodium acetate and isopropanol, washed with 70% ethanol. After drying, the genomic DNA pellet was dissolved using TE buffer (10 mM Tris-HCl pH 8.0, 1 mM EDTA). The purity of the extracted genomic DNA was measured using a BioSpec-nano spectrophotometer (Shimadzu Biotech, Kyoto, Japan), and the integrity was evaluated on a 1% (*w*/*v*) agarose gel.

Massive amplicon sequencing analysis of the gut microbiome DNA samples was processed and sequenced by Ebiogen, Inc. (Seoul, Korea). Briefly, the sequence library was prepared by the Illumina 16S Metagenomic Sequencing Library protocols. The targeted 16S rRNA genes were amplified by 16S V3–V4 primers; 16S Amplicon PCR Forward Primer with sequence of 5′TCGTCGGCAGCGTCAGATGTGTATAAGAGACAGCCTACGGGNGGCWGCAG3′ and 16S Amplicon PCR Reverse Primer with sequence of 5′GTCTCGTGGGCTCGGAGATGTGTATAAGAGACAGGACTACHVGGG-TATCTAATCC3′ [19]. The amplified products were normalized and pooled by PicoGreen. The size of libraries was verified by the TapeStation DNA screentape D1000 (Agilent, Santa Clara, CA, USA), followed by sequencing using the MiSeq™ platform (Illumina, San Diego, CA, USA). Amplicon errors were modeled in the merged fastq using DaDa2, filtering out noise sequence, correcting for errors in marginal sequences, removing chimeric sequences, removing singletons, and then de-replicating those sequences [20]. After removing the error sequences, the filtered 16S rDNA sequences (Appendix A) were used to identify individual microbes by matching the 16S rDNA sequences with the SILVA reference (region V3–V4) database (https://www.arb-silva.de/, accessed on 13 October 2021). The raw data were kept in the repository at Figshare (https://doi.org/10.6084/m9.figshare.17714834, accessed on 3 January 2022).

### 2.6. Data Analyses

All data and statistical analyses were determined as described previously [4,19]. In brief, the Q2-Feature classifier was used to classify bacterial species. The datasets after setting the denoise-single function as a default parameter were used. The q2-diversity under the option of “sampling-depth” was used for the calculation and statistical tests of diversity. A sequencing quality score threshold of at least 20 was used, and rarefaction depth was 11,510. The sequence similarity threshold for OTUs was 99%. After confirming the quality of sequencing results, the sequencing results were filtered by using the threshold values in QIIME2. OTU (operational taxonomic unit) and taxonomic classification tables were extracted into the phyloseq (1.28.0) package (R version 3.6.1) for the analysis of alpha and beta diversity. The permutational multivariate analysis of variance (PERMANOVA) in the vegan package in R was used to detect statistical differences in beta diversity metrics. ADONIS was used with 999 permutations (vegan package of R). All *p* values were corrected by Benjamini and Hochberg’s adjustment, and significance was declared at *p* < 0.05.

### 2.7. The α-Diversity Analysis for Relative Abundance Evaluation of Material and Microbiome

The α-diversity analysis for relative abundance was determined as described previously [4,19]. The α-diversity metrics for Shannon Index and abundance-based coverage (ACE) estimator were calculated using unfiltered values from the phyloseq package. The differences in alpha diversity and richness between groups were detected by core metrics analysis. Cumulative-sum scaling (CSS) normalization was implemented in the R package metagenomeSeq before being converted into relative abundance.

### 2.8. The β-Analysis for Relative Abundance Evaluation of Material and Microbiome

The β-diversity metrics were computed for non-metric multidimensional scaling (NMDS) from log-transformed OTU data using Bray-Curtis dissimilarity in the vegan package as described previously [4,19]. The NMDS was performed on the Bray-Curtis dissimilarity matrix using the metaMDS function in the vegan package, which reduced dimensionality while retaining as much information as possible about relationships among samples.

### 2.9. Establishment of Heatmap and Phylogenetic Tree

Heatmap and cluster analyses were performed using the relative abundances of all OTU values or core abundant OTU values in the Heatplus (2.30.0) package from Bioconductor and the vegan package in R as described previously [4,19]. Average linkage hierarchical clustering was used for cluster analysis. Bray–Curtis distance metrics were used for heatmap generation.

The phylogenetic tree for each sampling site was constructed as a raw sequence without any filtering to directly visualize sample richness associated with the relation to taxonomy classification as described previously [4,19]. The 16S rRNA sequences from the sampling site were aligned in ClustalW with default parameters. The results were aligned for use in establishing the maximum-likelihood phylogenetic trees (MEGAX). All phylogenetic trees were imaged in iTOL.

### 2.10. Co-Occurrence Network Construction

Co-abundance networks were produced by a permutation-renormalization-bootstrap network construction strategy as described previously [4,19] to observe the bacterial co-occurrence relationships through the mice’s weight change. All networks were independently established by splitting the OTU abundance matrix into 0m PBW, 0m MBW, 0m NBW, 3m PBW, 3m MBW, and 3m NBW groups. The microbial networks, links, and edges were created from OTU occurrence data. The multiple ensemble correlation protocol in CoNet, a Java Cytoscape plug-in, was used to identify significant co-presences across the samples while OTUs that occurred in fewer than three samples were deleted (“row_minocc” = 3). Five similarity indicators including Spearman and Pearson correlation coefficients, the mutual information score, and the Bray-Curtis and Kullback-Leibler dissimilarity were calculated by CoNet for the product of an ensemble network without a *p*-value merge. Significance was marked if *p* < 0.05 with Benjamini and Hochberg’s adjustment. If at least two of the five metrics indicated significant co-abundance between the two OTUs, the relationship was remained in the final network to be marked as an edge. A final co-occurrence network model was imaged using the Louvain algorithm to identify communities (igraph package).

### 2.11. Statistical Analysis and Quantification

The mice body characteristics were analyzed using one-way ANOVA represented as the mean ± SEM. The relative abundances of bacteria-containing feces were analyzed by the Mann-Whitney sum rank tests in R software. Significance was mentioned at *p* < 0.05 with Benjamini and Hochberg’s adjustment. All graphs were prepared using R software.

### 2.12. Ethics Approval

The animal protocols (KBSI-IACUC-21-29) were approved by the Institutional Animal Care and Use Committee of the Korea Basic Science Institute (KBSI), and all animal experiments were conducted in accordance with the Guide for the Care and Use of Laboratory Animals issued by the Laboratory Animal Resources Commission of that Institute.

## 3. Results

### 3.1. Random Replacement of the Gut Microbiome of Conventional Mice with the Subset of the Human Gut Microbiome Affected Body Weight Gain Pattern

The current dispute in intestinal bacteria’s regulation of the body weight of their host made us consider a different approach to pinpoint the bacteria responsible for regulating the body weight. Since subgroup analysis on large and complex data illuminates the culprit behind large and complex data, we applied this concept to an in vivo experiment. To identify the intestinal bacteria responsible for regulating body weight after a randomized subgroup analysis, the fresh fecal sample was fed to conventional mice, C57BL/6, for which the gut microbiome had already been removed by feeding with a mixture of 3 broad-spectrum antibiotics and nystatin (Figure 1A).

Replacement of their own original gut microbiome with the subsets of human gut microbiome affected the weights differently (Figure 1B,C). To evaluate the effect of the gut microbiome only on weight gain by excluding genetic factors, the body weight change was monitored in the same mice after replacing their gut microbiome. The weight changes of the mice during the 3 month experimental period were able to be grouped into three: the positive group (PBW; *n* = 5; body weight change = 5.9 ± 1.1 g), the unchanged group (MBW; *n* = 6; body weight change = 2.6 ± 0.1 g), and the negative group (NBW; *n* = 6; body weight change = −1.6 ± 0.8 g) (Figure 1B,C). In accordance with the weight change, the total body fat measured by CT scan was highest in PBW, medium in MBW, and lowest in NBW (Figure 1D and Appendix A). In addition, the histological examination confirmed a high degree of fat accumulation in more enlarged adipocytes in PBW, medium in MBW, and lowest in NBW (Figure 1E and Appendix A). The CT and histological examination results in Figure 1D,E indicate that fat accumulation was responsible for the body weight difference in the mice. The levels of blood glucose and lipid profiles did not change significantly before and after the gut microbiome replacement in the mice. However, blood pressure significantly decreased in NBW (Appendix A). Altogether, these results indicate that a different kind of subset of the human gut microbiome randomly replaced the original gut microbiome in each mouse so that the different subset affected the body weight differently.

### 3.2. Different Kinds of Gut Microbiome Were Replaced in Each of the Experimental Mice

The differential effects on the body weight gain after the gut microbiome replacement prompted us to compare changes in the composition of the gut microbiome before and after gut microbiome replacement in mice. For all identified 16S rDNA sequences, operating taxonomic units (OTUs) of the gut microbiomes of the fecal samples were able to be classified into nine different phyla: Bacteroidetes (53.89%), Firmicutes (37.47%), Verrucomicrobia (4.94%), Patescibacteria (1.42%), Proteobacteria (1.33%), Actinobacteria (0.45%), Tenericutes (0.308%), Cyanobacteria (0.189%), and Lentisphaerae (0.001%) (Appendix A). After classification of the OTUs at phylum level, the OTUs were further classified until species level.

Comparison of the OTUs visualized a clear before and after difference in all of the mice (Figure 2, Figure 3, Appendix A), indicating that the replacement of their original gut microbiomes with those of the humans was successful. The shift in gut microbiome was noticeable at large classification levels. The original Firmicutes/Bacteroidetes ratio, which was 0.91, decreased to 0.62 after the gut microbiome shift in the PBW group (Figure 2A). In accordance with the PBW group, the Firmicutes/Bacteroidetes ratio increased from 0.58 to 0.8 in the NBW group, while the ratio was not changed after gut microbiome replacement in the MBW group. Overall, the taxonomic data showed that body weight gaining was negatively associated with the Firmicutes/Bacteroidetes ratio, suggesting that Firmicutes plays an anti-obesity role while Bacteriodes plays a pro-obesity role. Despite noticeable differences at large taxonomic levels, the differences at genus and species levels were more notable (Figure 2E and Figure 3B).

### 3.3. The Differences in Body Weight Gain Were Associated with the Different Shifts in the Composition of the Gut Microbiome

After confirming the individual differences in the composition of the replaced gut microbiome and subsequent body weight gain, the gut microbiome of the individual mice was analyzed. The richness (ACE and Fisher) and evenness (Shannon, Evenness, Simpson, and InvSimpson) measurement of α-diversity metrics showed that different subsets of the human gut microbiome were replaced in each group of mice (Figure 4 and Appendix A). Although the α-diversity indices considering both richness and evenness showed slight differences between the three groups, the separate measurement of richness and evenness visualized the structural difference in the ecological community. The evenness indices before and after the gut microbiome replacement were similar, but the richness indices were increased after the gut microbiome replacement, which suggests that the human gut microbiome is more diverse, although the compositional characteristics of the human and mouse gut microbiomes are similar. In addition, the richness and diversity indices of the mice that gained high body weight were slightly lower than the rest of the two groups, indicating less diversity in the gut microbiome of PBW.

Other than α-diversity analyses, β-diversity analyses also confirmed that the replaced gut microbiome was much more diverse than its original gut microbiome in each mouse. As shown in Figure 5A,B, both β-diversity metrics measured by NMDS and MDS plots showed more diverse features after the gut microbiome replacement.

To investigate the effect of the gut microbiome on the body weight gain, we compared the composition of the gut microbiome at the moment of before (T0) and 3 months after the replacement (T3) of the gut microbiome. The comparison of the unsupervised hierarchical clustering of the top-ranked OTUs based on the Bray-Curtis distance between T0 and T3 confirmed that replacement of the gut microbiome affected the three groups of mice differently. The unsupervised hierarchical cluster analysis at T0 showed that each of the three groups of mice was completely unrelated to each other (Figure 5C, left). However, the mice clustered according to the weight gain groups (Figure 5F, right). This study considered only body weight changes in the same individual, and thus the gut microbiome at the starting point did not show any group difference because the starting points were not related to each other. However, since the gut microbiome affected the body weight gain of the mice, the unsupervised hierarchical cluster analysis at T3, which reflected the composition of the gut microbiome, grouped the individual mice according to differences in body weight gain.

### 3.4. Different Kinds of Microbial Communities Were Established among the Three Groups of Mice

The gross differences between the gut microbiomes of the PBW, MBW, and NBW groups were analyzed by constructing phylogenetic trees. A maximum-likelihood phylogeny of each microbiome of the three groups was built on the basis of the 16S rDNA sequence (Figure 6). The gut microbiomes of all of the three groups expanded their diversity after the replacement of the gut microbiome: 50.7% in PBW, 72.1% in MBW, and 74.8% in NBW at the species level (Appendix A). Other than the increased diversity, the intestinal bacteria constituting the gut microbiome were changed dramatically after the replacement. The main bacteria constituting their original gut microbiomes were Bacteroidetes, Firmicutes, and Patescibacteria at a phylum level; Bacteroidia, Clostridia, and Saccharimonadia at the class level; Bacteroidales, Clostridiales, and Saccharimonadales at the order level; and Muribaculaceae, Lachnospiraceae, and Ruminococcaceae at the family level (Appendix A). Although these bacteria still dominated in the gut microbiome after antibiotics treatment and subsequent fecal feeding, compositional changes in the bacteria were noted, suggesting establishment of different kinds of microbial communities. These data suggest that the human gut microbiome is not only more diverse but also its composition differs significantly from that of mice.

The phylogenetic analysis showed that the replaced gut microbiomes of PBW, MBW, and NBW groups were different, although their original gut microbiomes did not differ from each other (Figure 6). Consistent with the phylogenic analysis, co-occurrence network analysis also showed that the microbial communities became more diverse after replacement of the gut microbiome in all three of the classified groups (Figure 7 and Appendix A). The 34 communities of PBW before gut microbiome replacement were expanded to 45 while the 37 communities became 30 in MBW and the 33 communities became 47 in NBW. Unlike numbers of communities, there was not any specialty in the nodes and edges within the microbial communities, suggesting that bacteria interact with each other in a similar way. The fewer microbial communities in PBW compared to MBW and PBW indicates that body weight gain is related to a less diverse gut microbiome.

### 3.5. Intestinal Microbes That Affect Body Weight Gain Were Identified at the Species Level

The above gross gut microbiome analyses show the association between gut microbiome composition and body weight gain. However, the group analyses do not show a group of bacteria responsible for pro-body weight gain and anti-body weight gain (Figure 2, Figure 3, Figure 4, Figure 5, Figure 6 and Figure 7). This suggests that the regulation of body weight gain by the gut microbiome might occur by individual bacteria rather than as a group-wide phenomenon such as Firmicutes or Bacteroidetes, contrary to the previous reports [7,21].

Although the group analyses did not show a group of intestinal bacteria associated with body weight gaining, a comparison at the species level showed intestinal bacteria that are associated with body weight gain (Figure 3 and Appendix A). Among the bacterial species, uncultured bacterium (NR_074436.1), uncultured bacterium (NR_144750.1), and uncultured bacterium (NR_0421101.1) were most abundant in PBW, indicating the positive association of these bacteria with body weight gain. Meanwhile, *Trichinella pseudospiralis*, uncultured bacterium (NR_024815.1), and uncultured bacterium (NR_144616.1) were most abundant in NBW, indicating the negative association of these bacteria with body weight gain. The taxonomic classification of these bacteria was shown in Appendix A. These uncultured bacteria associated with positive body weight gain belonged to Verrucomicrobia, Bacteroidetes, and Firmicutes, while the bacteria associated with negative body weight gain belonged to two Bacteroidetes and one Firmicutes.

## 4. Discussion

The gut microbiome plays an essential role in the energy extraction by its host from food by fermentation processes [22] and an increase in villous vascularization [23]. It also activates the storage of triglycerides in the adipose tissue and liver by inhibiting lipase [23]. Because the gut microbiome regulates the energy homeostasis of its host so significantly, it is not surprising to find the obvious critical contribution of the gut microbiome in body weight gain. Gut microbiome transplantation into germ-free mice dramatically increased the body weight up to 19.1% [24]. Especially, the germ-free mice colonized with the gut microbiome of an obese donor gained 19.7% more body weight than when colonized with those of a lean donor [25]. In addition, the obesity in mice was transmissible by transplanting the gut microbiota of conventional obese mice to normal-weight, germ-free animals [26].

Despite the obvious and critical role of the gut microbiome in the contribution to body weight gain, the bacteria contributing to body weight have not been defined. Firmicutes and Bacteroidetes are the two major phyla in feces. Original research that studied the association between obesity and the gut microbiome showed that Firmicutes positively contributes to obesity while Bacteroidetes contributes negatively [20]. This result was supported by several different works on gut microbiome analysis in animal and human studies, proposing the Firmicutes/Bacteroidetes ratio as a possible hallmark of obesity [9,27,28,29]. However, contrary to these results, many studies did not observe a positive association between the Firmicutes/Bacteroidetes ratio and obesity, but rather the Firmicutes/Bacteroidetes ratio was negatively associated with obesity [30,31]. An interrogated reanalysis using published data showed that the Firmicutes/Bacteroidetes ratio is not associated with obesity [32]. Meta-analysis also did not support the positive correlation between the Firmicutes/Bacteroidetes ratio and obesity [33]. In accordance with more recent works, our data show that body weight gaining was negatively associated with the Firmicutes/Bacteroidetes ratio (Figure 2A). Considering recent technological advances in metagenome sequencing, it would be reasonable to speculate that recent works would be more accurate than the earlier works. Although this work coincides with the recent finding of a negative association of the Firmicutes/Bacteroidetes ratio with obesity, the effect of the gut microbiome on body weight gaining at large taxonomic levels was quite ambiguous (Figure 4, Figure 5, Figure 6 and Figure 7).

Besides confirmation of recent findings on the negative association of the Firmicutes/Bacteroidetes ratio with obesity by using an animal experiment, this work pinpointed the intestinal microbes associated with body weight gaining at low taxonomic levels. Because of the functional heterogeneity of individual bacteria within the same phylum, investigation of the gut microbiome at low taxonomic levels has not generated statistically meaningful data at lower taxonomic levels [27,28,29,30,31,32,33]. The reason behind the lack of findings at low taxonomic levels in gut microbiome research would be due to host factors. It is obvious that host factors such as genetic background may play more or at least similarly significant roles in determining the phenotypes of a host than the gut microbiome does. Therefore, exclusion of host factors would generate the most reliable analysis result of gut microbiome function. In this work, we excluded the host factors by comparative analysis of before and after gut microbiome modification within the same individual. Because the different sets of gut microbiomes were compared within the same individual following the phenotype change, we were able to find specific bacteria regulating body weight gain (Figure 3 and Appendix A).

In this work, a minimal amount of fecal samples from overweight individuals were inoculated to the experimental mice, for which the gut microbiome was disrupted by feeding with an antibiotic and antifungal complex. The gut microbiome of overweight individuals would be more diverse than that of obese individuals in terms of the bacterial compositions associated with body weight gaining, and a small sample number cannot represent the entire population. Thus, if a minimal amount of fecal sample is inoculated into the mice depleting gut microbiome, a different gut microbiome would be established in each of the individual mice. In fact, this work showed that a completely different gut microbiome was established on an individual basis by feeding a minimal amount of fecal sample to the mice and depleting the gut microbiome (Figure 2, Figure 3, Figure 4, Figure 5, Figure 6 and Figure 7). Because a different gut microbiome was established in each of the individual mice, the phenotype (body weight gaining) of each of the mice was dramatically changed after replacing the gut microbiome (Figure 1). The dramatic change in phenotype after gut microbiome replacement within the same individuals clearly proved the significance of the gut microbiome in body weight gaining.

Finally, this work successfully showed that replacement of the gut microbiome within the same individual is one approach to analyze a gut microbiome consisting of a very heterogeneous, complex population of trillions of microbes by excluding the host factors. We believe that this concept could be applied to identify the intestinal microbes associated with other various phenotypes or diseases in humans.

## Figures and Tables

**Figure 1 microorganisms-10-01062-f001:**
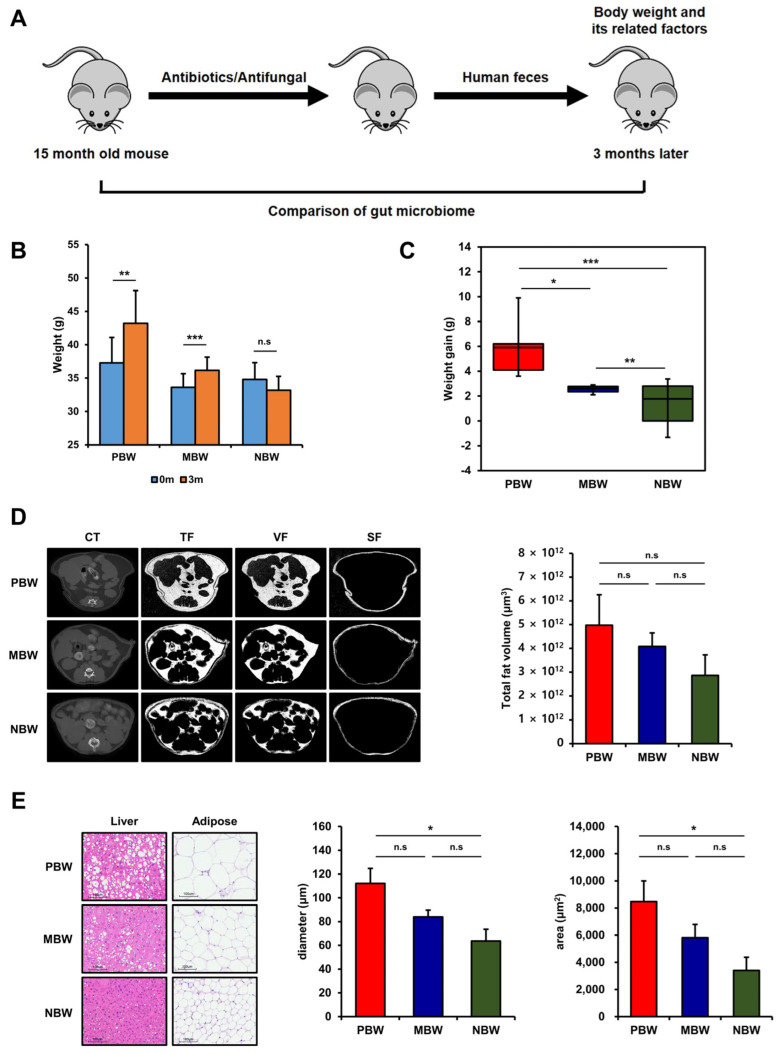
Grouping of mice based on body weight change before and after the intestinal microbiota transplantation. (**A**) The schematic diagram of the experimental strategy. (**B**) The body weight of the experimental mice before and after the gut microbiota replacement. (**C**) The body weight change of the experimental mice between before and after the gut microbiota replacement. (**D**) The representative micro-CT image of PBW, MBW, and NBW used for abdominal fat analysis. (**E**) The representative histological image of the H&E-stained liver and adipose tissue (scale bar, 100 μm) of PBW, MBW, and NBW. The values represent the mean ± SEM. TF, total fat; VF, visceral fat; SF, subcutaneous fat. * *p* < 0.05; ** *p* < 0.01; *** *p* < 0.001; n.s: not significant (*p* > 0.05).

**Figure 2 microorganisms-10-01062-f002:**
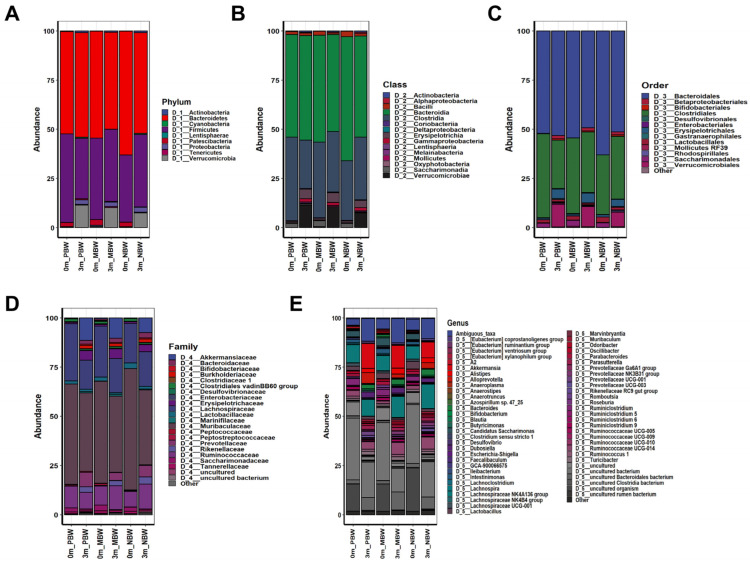
The gut microbiome composition before and after gut microbiome replacement in the experimental mice. The relative compositional changes in the gut microbiome at the (**A**) phylum, (**B**) class, (**C**) order, (**D**) family, and (**E**) genus levels. For the abundance values of all groups, the average abundance values of individuals belonging to each group were used.

**Figure 3 microorganisms-10-01062-f003:**
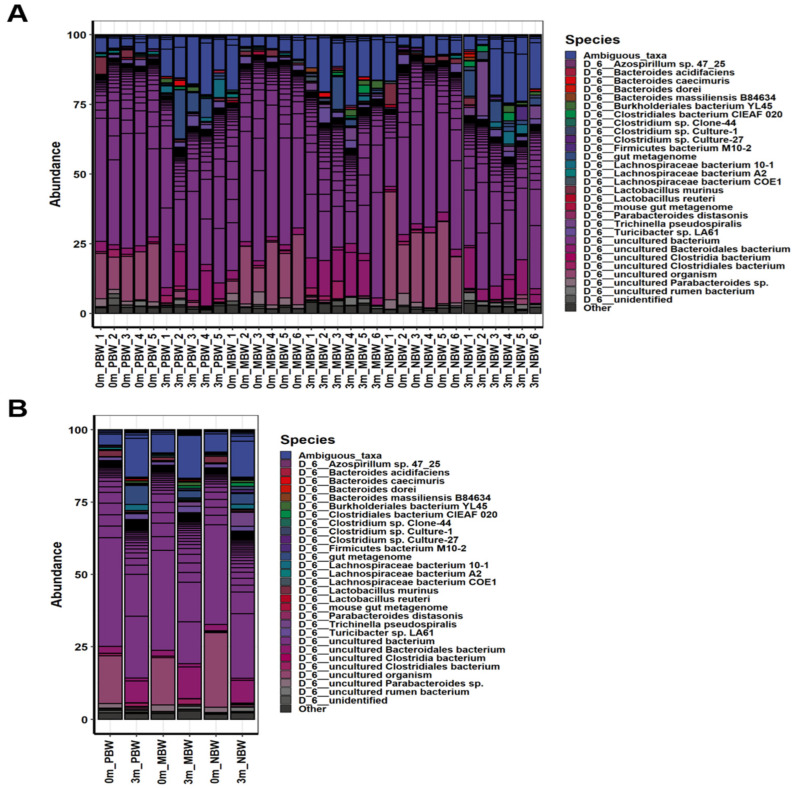
Compositional changes in the gut microbiome before and after the gut microbiome replacement. The relative compositional changes before and after the replacement in the gut microbiome at the species level for PBW, MBW, and NBW groups are shown. (**A**) Abundance values of individuals belonging to each group. (**B**) The average abundance values of each group.

**Figure 4 microorganisms-10-01062-f004:**
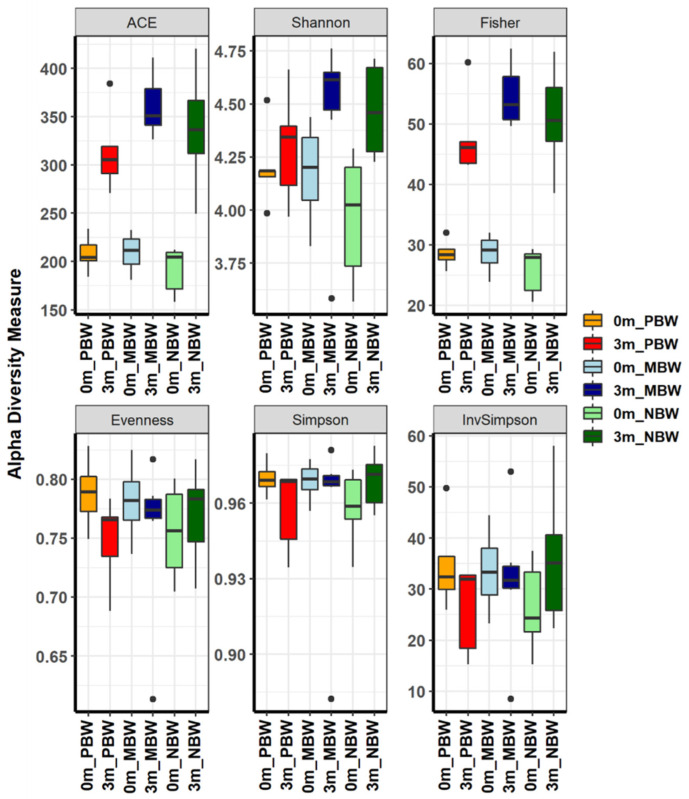
The α-diversity indices of the gut microbiome of the PBW, MBW, and NBW groups. Species richness and diversity calculated by ACE richness, Shannon diversity and Fisher’s alpha, evenness, Simpson, and inverse Simpson for the PBW, MBW, and NBW groups are shown for before and after the gut microbiome replacement.

**Figure 5 microorganisms-10-01062-f005:**
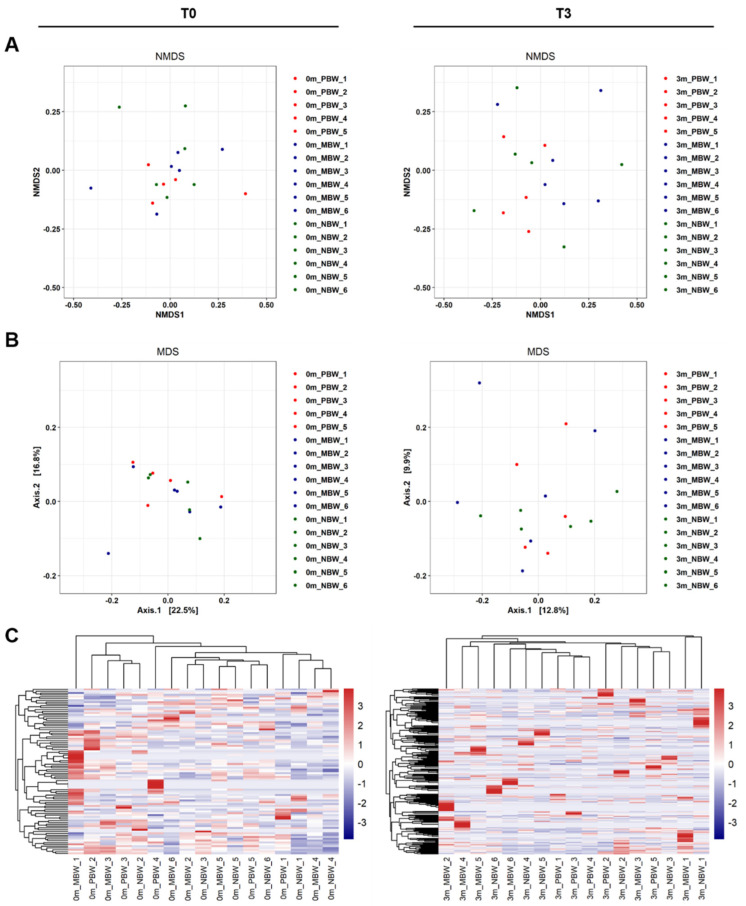
The β-diversity comparison of the gut microbiome of the PBW, MBW, and NBW groups. (**A**) NMDS plots showing differences in the gut microbiome before (T0, (**left**)) and after (T3, (**right**)) the gut microbiome replacement in the PBW, MBW, and NBW groups based on Bray-Curtis distances using OTUs. (**B**) MDS plots showing difference in the gut microbiome before (T0, (**left**)) and after (T3, (**right**)) the replacement in the PBW, MBW, and NBW groups based on Bray-Curtis distances using OTUs. (**C**) Heatmaps of the microbial composition of the PBW, MBW, and NBW groups before (T0, (**left**)) and after (T3, (**right**)) the replacement based on Bray-Curtis distance matrix measured at the species level.

**Figure 6 microorganisms-10-01062-f006:**
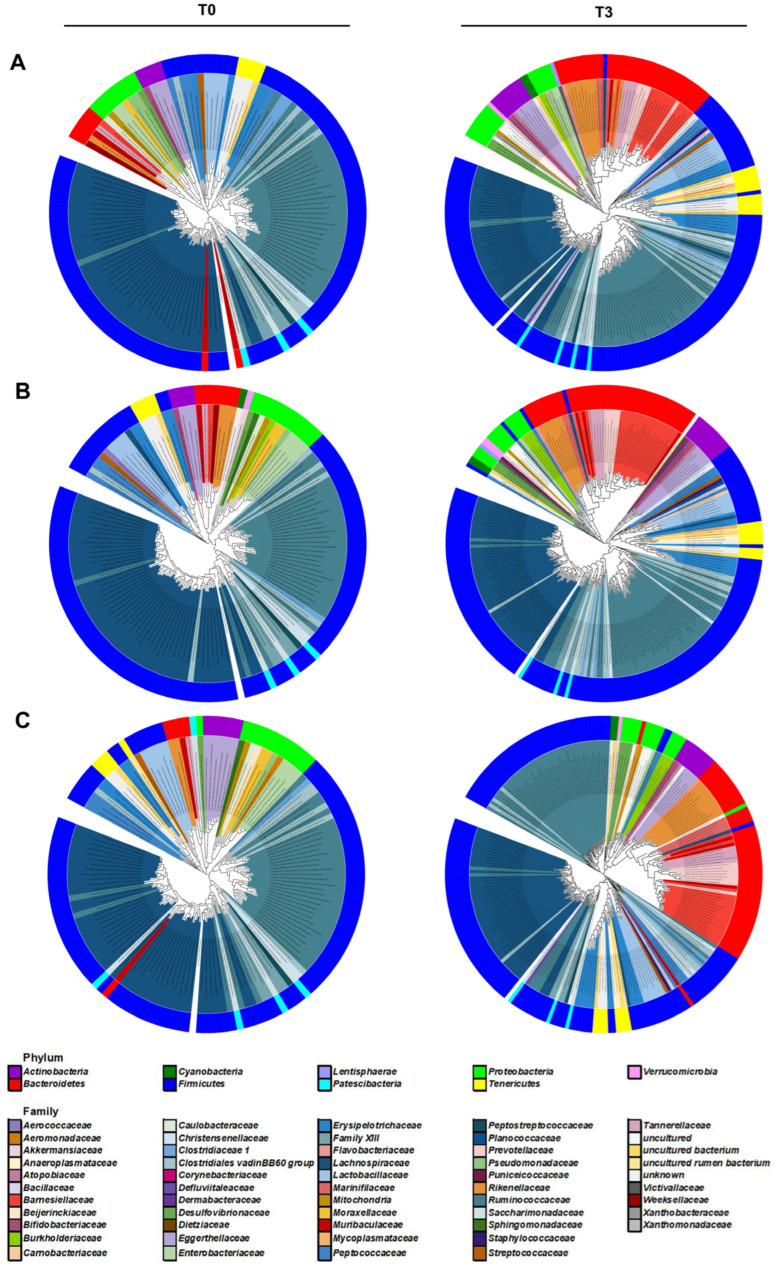
The maximum-likelihood phylogenetic tree consisting of the gut microbiome taxa in the PBW, MBW, and NBW groups. The outer rings of the circular dendrogram represent the phylum level, and the inner layers represent the family level. (**A**–**C**) are PBW, MBW, and NBW groups each before (T0, (**left**)) and after (T3, (**right**)) the replacement.

**Figure 7 microorganisms-10-01062-f007:**
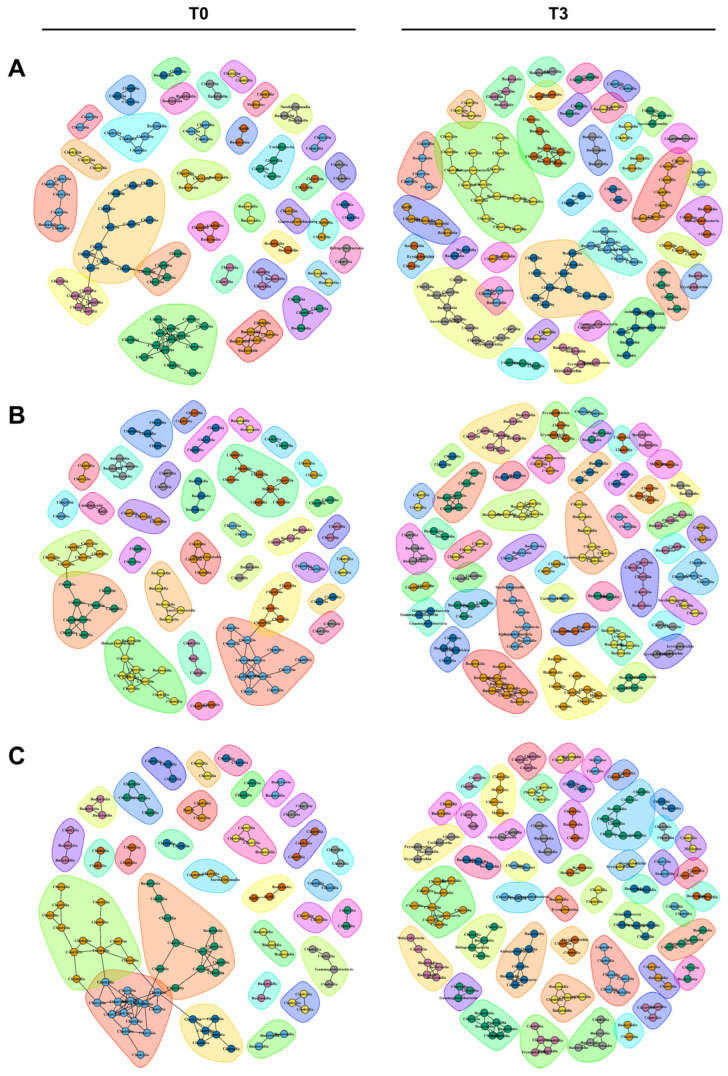
The co-occurrence network analysis by the ReBoot algorithm for the PBW, MBW, and NBW groups. The color-coded network graphs indicate the co-occurring and mutually exclusive interactions between OTUs. Black letters in the nodes correspond to the class level of the OTUs. Transparent shapes represent the network communities as determined by the Louvain modularity algorithm. (**A**–**C**) are PBW, MBW, and NBW groups before (T0, (**left**)) and after the replacement (T3, (**right**)) each.

## Data Availability

The raw data were deposited in the repository at Figshare (https://doi.org/10.6084/m9.figshare.17714834, accessed on 3 January 2022).

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
