# Peer review of "Comparative Analysis of Original and Replaced Gut Microbiomes within Same Individuals Identified the Intestinal Microbes Associated with Weight Gaining"

_microorganisms, 2022, doi:10.3390/microorganisms10051062_

Round 1

Reviewer 1 Report

Comparative analysis of original and replaced gut microbiomes 2 within same individuals identified the intestinal microbes as-3 sociated with weight gaining, it is interesting manuscript, but this manuscript need to be improved.

Major concerns

  1. The fecal from 10 overweight volunteers, please illustrated the characteristic of volunteers’ BMI? Weight? Gender? Compositions of gut microbiota ?And how to collect and store the fecal sample
  2. Line 89-92, pleased clarified the definition of PBW, MBW, and NBW, how many mice are in each group?
  3. Line 253-259 could be integrated into the section of 2.5 16S rRNA Gene Sequencing
  4. 11. Statistical Analysis and Quantification, the statistic is for gut microbiota, but other experiments such as Figure 1, which statistic that author used?
  5. Line 282-283, which result indicates richness and which result indicates evenness?
  6. In figure 4, why PBW group has 5 mice and the other two group mice have 6 mice?
  7. The results of gut need more statistic methods to prove that which microoganisms is related to the body weight

Minor concerns:

  1. Line 81, please showed the whole name of JMS medium.

Author Response

Major concerns

1. COMMENT:

The fecal from 10 overweight volunteers, please illustrated the characteristic of volunteers’ BMI? Weight? Gender? Compositions of gut microbiota? And how to collect and store the fecal sample

Response to Comment:

According to your comment, we described the procedure and methods in detail. Please refer to Line 82 - 83. Thank you.

2. COMMENT:

Line 89-92, pleased clarified the definition of PBW, MBW, and NBW, how many mice are in each group?

Response to Comment:

According to your comment, we described the procedure and methods in detail. Please refer to Line 94 - 99. Thank you.

3. COMMENT:

Line 253-259 could be integrated into the section of 2.5 16S rRNA Gene Sequencing

Response to Comment:

According to your suggestion, we described the procedure and methods in detail. Please refer to section 2.5. Please refer to Line 150 - 153.Thank you.

4. COMMENT:

Statistical Analysis and Quantification, the statistic is for gut microbiota, but other experiments such as Figure 1, which statistic that author used?

Response to Comment:

Figures 1B, 1D, and 1E were analyzed by using one-way ANOVA. According to your comment, we added description about statistical analysis on the statistical analysis method in Figure 1 legend as well as section 2.11. Please refer to Figure 1 legend and section 2.11. Thank you.

5. COMMENT:

Line 282-283, which result indicates richness and which result indicates evenness?

Response to Comment:

According to your comment, we clarified the richness evenness of the result data. Please refer to Line 286 - 287. Thank you.

6. COMMENT:

In figure 4, why PBW group has 5 mice and the other two group mice have 6 mice?

Response to Comment:

After shifting the original gut microbiome of mice to new ones, we observed the body weight change. Since the experimental mice were grouped based on the degree of the weight change, the number of mice per group were different each other.

7. COMMENT:

The results of gut need more statistic methods to prove that which microorganisms is related to the body weight

Response to Comment:

We added the new statistical analyses following your suggestion. Please refer to Figure S7. Thank you.

Minor concerns:

  1. COMMENT:

Line 81, please showed the whole name of JMS medium.

Response to Comment:

JMS medium is a culture medium supporting universal growth of intestinal microbes. We recently submitted a paper about JMS medium to Nature. This paper is in review state. According to your comment, we added citation in Line 83 and 494. Thank you.

Reviewer 2 Report

The study is focused on faecal transplantation in mice to study microbiome impact on body weight. Although several approaches have been published in this way, the goal is still interesting for society, health and wellness. However, this study lacks on structural concepts, such as controls. Finally, the results obtained are not robust enough to be extrapolated in the conclusions claimed. A set of major and minor comments are described.

Major

Authors also need to consider that replacement is made in a non-natural host. That is, they are not replacing an obese mouse microbiome in a lean mouse, but a human microbiome in a mouse. Differences in microbiomes are detected in different animals, and how they metabolize nutrients is dependent.

The main conclusion of this work is at least questionable. The unidentified groups may or may not be associated with one species, or several, the authors are blind on this point and cannot extrapolate from these results.

L69: please include “in our study”. These results can’t be extrapolated to the whole obesity process

L80: can authors include BMC and any characterization or definition of metabolic and weight status of the volunteers?

Did the authors apply for ethical committee for human samples? did the volunteers have a written consent?

Do the authors consider that the antibiotic treatment was similar in all the animals? Did they test (either in this trial or previously in others) that this is equally efficient and thus they can consider an absence of microbiome? If it is not the case, one could consider that, additionally to the host characteristics (in the same breed) differences in the initial microbiome could be affecting the microbiota development

Do the authors have a figure or image where rarefraction curves ae shown? Do they achieve the plateau?

In microbiome analysis, authors sequenced and compared microbiome from mice before antibiotic treatment, and after 3 months of microbiome replacement. If authors consider that the antibiotic treatment has been successful, the comparison with the initial microbiome has no sense, and it does not apport any relevant data to the study. If it can’t be assumed, another point of microbiome study should be included (or just after the antibiotic treatment to ensure what kind of bacteria is still present, or/and a few days after replacement, to have an initial picture of what is the microbiome established initially as starting point). As authors do not have any, comparison turn more difficult, and also conclusions derived. Moreover, have the authors sequenced the fresh pool microbiome used for the replacement? This could help to really understand the level of replacement.

In beta-diversity, can the authors run any linear mixed model to  study how the individual, and weight, and their interaction, affect and if this is significant? Moreover, in material and methods, it is explained PERMANOVA analysis for beta-diversity, but this is not included in Results. Can the authors clarify if it was done, and  the results obtained?

L330-353: please, summarize the main findings, and make this section more concise, its very hard to follow it. If all this data (%)  is included in the files, it is not necessary to writhe again in the text

It could be convenient to make a DESEq2 study, time dependent, to see specific variations in taxa in the three groups. Also, there are tools for correlating specific parameters with microbiome profiles, as MaAslin (the fact that the authors have not analysed it, does not imply that there are not)

L370*-372. The study lacks on controls and power (number of animals, subjects, ect) to conclude this. The results are not robust neither consistent enough.

L33-436 the trial done and the results do not substantiate the conclusion included in this paragraph. The study is based on one pool (not sequenced), with 5-6 mice each, without controls, and transplanting human microbiome in a host that possesses a different microbiome and different functional taxa groups of metabolizing nutrients, and also produce molecules such as SCFAs, vitamins, etc.

MInor

L14: please correct to “mechanisms of action”

L24: please change to “ Also, body weight gain was not determined by…”           

L35-36: What do the authors want to say with this sentence? Failure in positive interactions? There are always functional interactions, as gut microbiome confers functions to the host, which not always are positive or beneficial. Please rewrite for a more accurate information.

L54-56: moreover, there are other considerations, as in the case of microbiome, how the samples are extracted, sequenced (primers and the area they cover) and analyzed, dramatically influence the results obtained.

L57 : please correct “are determined by the interaction between “

L58: Please correct me, but in your cited paper (Lkhagva et al. 2021), the analysis was focused on the factors affecting gut microbiome, not on how microbiome affects phenotype factors so the statement is not supported by this cite. Please, either introduce another work that supports the text, or change the text with a focus on microbiome.

L59  “please include, “Even defined genetic backgrounds, the effect of the host genes…. “

L89-93: please include exact information weight gain/loss ± SD for each group

L131: the authors performed amplicon-based sequencing, not metagenome sequence. Please correct to “Massive amplicon sequencing analysis…”

L132: please delete “a Korean commercial company” an substitute per “(city, Korea)” as an outsourced service

L134: are the primers being cited elsewhere initially? If yes, please include the information

L152: is the “table.qzv” include in any supplementary data? If not, delete this part

L152: what was the sequence similarity threshold  for OTUS?  97%? Please include. For the future, please revise ASVs approach, that is more convenient for collecting the heterogeneity of a microbiome

L157; please correct “All P values were corrected…”

M&M: Please include for all the software’s and pipelines used their respective cites.

L179: did the authors mean “raw”? if yes, please correct

L182 Please correct “were aligned”

L187: this is not for microbial, this is for bacterial…

L208 Please correct “The animal protocols were approved…”, or The animals protocol was approved, deeding on what the authors do want to explain

L215-218: the initial text is more related with discussion or introduction. Results must  be short and concise.

L253-260. This must be included in material and methods section, not in results. Please change

L307 please correct “the mice clustered according to the weight gain groups”

L367: please specify why is obvious

Please do not consider the term “up or down regulated” this is not transcriptomics.

Author Response

Major

1. COMMENT:

Authors also need to consider that replacement is made in a non-natural host. That is, they are not replacing an obese mouse microbiome in a lean mouse, but a human microbiome in a mouse. Differences in microbiomes are detected in different animals, and how they metabolize nutrients is dependent.

Response to Comment:

We agree with you in that replacement of gut microbiome was made in a non-natural host. Of cause, some bacteria may affect the phenotype or diseases of their hosts differently. However, most bacteria affect their host similar ways.

The purpose of this work was to evaluate only effect of gut microbiome after excluding the genetic factors of hosts. Therefore, we compared two different gut microbiomes within a same individual mouse. If microbiome originated from different sources are used, the effect of microbiome would be clearer. Because of this reason, we used human gut microbiome to replace original microbiome of mice.

2. COMMENT:

The main conclusion of this work is at least questionable. The unidentified groups may or may not be associated with one species, or several, the authors are blind on this point and cannot extrapolate from these results.

Response to Comment:

According to your concern, we added new statistical method. Thank you.

Original research on gut microbiome suggested that body weight gain was determined by a group of bacteria such as Firmicutes, Bacteroidetes, etc. However, recent works are consistently showing that body weight gain was not determined by a group of bacteria, rather body weight gain is regulated at species level, as we described in the manuscript. This work clearly showed that specific species, not a group of bacteria, regulate the body weight gain.

To further validate the regulation of body weight gain at species level not by a whole taxonomic group according to your concern, we added more statistical analyses. Please refer to newly added statistical analyses of Figure S4, Figure S6, and Figure S7. Thank you.

3. COMMENT:

L69: please include “in our study”. These results can’t be extrapolated to the whole obesity process

Response to Comment:

We included “in our study” in the line. Thank you.

4. COMMENT:

L80: can authors include BMC and any characterization or definition of metabolic and weight status of the volunteers?

Response to Comment:

According to your comment, we described the procedure and methods in detail. Please refer to Line 82 - 86. Thank you.

5. COMMENT:

Did the authors apply for ethical committee for human samples? Did the volunteers have a written consent?

Response to Comment:

We consulted with the ethical committee of our university before starting this work. According to university committee, since this work was for just use of fecal sample, an approval by the ethical committee was not necessary in this research. Thank you.

6. COMMENT:

Do the authors consider that the antibiotic treatment was similar in all the animals? Did they test (either in this trial or previously in others) that this is equally efficient and thus they can consider an absence of microbiome? If it is not the case, one could consider that, additionally to the host characteristics (in the same breed) differences in the initial microbiome could be affecting the microbiota development

Response to Comment:

Antibiotics work on bacteria, not on the host. Therefore, the anti-bacterial efficacy of antibiotics depends on bacteria. Furthermore, the reason for using antibiotics in this experiment is to temporarily disrupt the intestinal microbial community of mice to help newly introduced human intestinal microorganisms settle well. As we did not aim for the complete absence of a microbiome, it was not necessary to achieve the same effect in all mice. Thank you.

7. COMMENT:

Do the authors have a figure or image where rarefaction curves ae shown? Do they achieve the plateau?

Response to Comment:

According to your comment, we added rarefaction curves in Figure S6. As you can see, the rarefaction curves achieved plateau. Thank you.

8. COMMENT:

In microbiome analysis, authors sequenced and compared microbiome from mice before antibiotic treatment, and after 3 months of microbiome replacement. If authors consider that the antibiotic treatment has been successful the comparison with the initial microbiome has no sense, and it does not apport any relevant data to the study. If it can’t be assumed, another point of microbiome study should be included (or just after the antibiotic treatment to ensure what kind of bacteria is still present, or/and a few days after replacement, to have an initial picture of what is the microbiome established initially as starting point). As authors do not have any, comparison turn more difficult, and also conclusions derived. Moreover, have the authors sequenced the fresh pool microbiome used for the replacement? This could help to really understand the level of replacement.

Response to Comment:

As we described in the manuscript, the purpose of this research is to analyze two different gut microbiomes within a same individual after replacing gut micro-biome with a new one to exclude the host factors. We did not aim to replace the original gut microbiome with a specific microbiome. Rather, we targeted to disrupt the intestinal microbial community of mice by feeding antibiotics to help newly introduced human intestinal microorganisms settle well. As you can see, metagenome sequencing showed that the gut microbiome has been changed completely. Thank you.

9. COMMENT:

In beta-diversity, can the authors run any linear mixed model to study how the individual, and weight, and their interaction, affect and if this is significant? Moreover, in material and methods, it is explained PERMANOVA analysis for beta-diversity, but this is not included in Results. Can the authors clarify if it was done, and the results obtained?

Response to Comment:

Based on our experimental design, it is not possible to apply a linear mixed model in this beta-diversity analysis. The results of PERMANOVA analysis for beta-diversity, i.e. non-metric multidimensional scaling (NMDS), is shown in Fig 4A and 4B. Thank you.

10. COMMENT:

L330-353: please, summarize the main findings, and make this section more concise, it’s very hard to follow it. If all this data (%) is included in the files, it is not necessary to write again in the text.

Response to Comment:

We summarized the description. Please refer to the change in L335-342. Thank you.

11. COMMENT:

It could be convenient to make a DESEq2 study, time dependent, to see specific variations in taxa in the three groups. Also, there are tools for correlating specific parameters with microbiome profiles, as MaAslin (the fact that the authors have not analysed it, does not imply that there are not)

Response to Comment:

We added the new analyses following your suggestion. Please refer to Figure S4. Thank you.

12. COMMENT:

L370*-372. The study lacks on controls and power (number of animals, subjects, etc) to conclude this. The results are not robust neither consistent enough.

Response to Comment:

The number of experimental animals depends on experimental design. Some experiments require hundreds of animals, while, in other experiments, 3~5 animals are enough because it can achieve a statistical significance. Currently, animal care committees in most countries force to reduce animal number if possible.

As you can see, we achieved enough statistical significance in this work. Thank you.

13. COMMENT:

L433-436 the trial done and the results do not substantiate the conclusion included in this paragraph. The study is based on one pool (not sequenced), with 5-6 mice each, without controls, and transplanting human microbiome in a host that possesses a different microbiome and different functional taxa groups of metabolizing nutrients, and also produce molecules such as SCFAs, vitamins, etc.

Response to Comment:

This work is to analyze the effect of gut microbiome with a same individual after excluding the genetic factors of hosts. Since two different gut microbiomes within a same individual mouse were compared in this work, the mice between before and after replacement of gut microbiome were control each other. Thank you.

Minor

1. COMMENT:

L14: please correct to “mechanisms of action”

Response to Comment:

We corrected it as you advised. Please refer to line 14.

2. COMMENT:

L24: please change to “Also, body weight gain was not determined by…”

Response to Comment:

We corrected it as you advised. Please refer to line 24.

3. COMMENT:

L35-36: What do the authors want to say with this sentence? Failure in positive interactions? There are always functional interactions, as gut microbiome confers functions to the host, which not always are positive or beneficial. Please rewrite for a more accurate information.

Response to Comment:

The meaning of functional interaction is different from that of positive interactions. There are various non-functional interactions which do not provide any function to host.

4. COMMENT:

L54-56: moreover, there are other considerations, as in the case of microbiome, how the samples are extracted, sequenced (primers are the area they cover) and analysed, dramatically influence the results obtained.

Response to Comment:

Of cause, these factors can affect analysis results. However, the sentence is to describe the feature of the complexity of interaction between gut microbiome and the genetic factors of hosts, not experimental limitations or errors.

5. COMMENT:

L57: please correct “are determined by the interaction between”

Response to Comment:

We corrected it as you advised. Please refer to line 58.

6. COMMENT:

L58: Please correct me, but in your cited paper (Lkhagva et al. 2021), the analysis was focused on the factors affecting gut microbiome, not on how microbiome affects phenotype factors so the statement is not supported by this cite. Please, either introduce another work that supports the text, or change the text with a focus on microbiome.

Response to Comment:

We have removed reference 4 (Lkhagva et al, 2021) from this section.

7. COMMENT:

L59: “please include, “Even defined genetic backgrounds, the effect of the host genes…”

Response to Comment:

We corrected it as you advised. Please refer to line 60-61.

8. COMMENT:

L89-93: please include exact information weight gain/loss ± SD for each group

Response to Comment:

In this paper, the statistical values are unified with ± SEM, so I calculated ± SEM and added it to the paper. Please refer to line 94-99.

9. COMMENT:

L131: the authors performed amplicon-based sequencing, not metagenome sequence. Please correct to “Massive amplicon sequencing analysis…”

Response to Comment:

We corrected it as you advised. Please refer to line 138.

10. COMMENT:

L132: please delete “a Korean commercial cmpany” an substitute per “(city, Korea)” as an outsourced service

Response to Comment:

We corrected it as you advised. Please refer to line 139.

11. COMMENT:

L134” are the primers being cited elsewhere initially? If yes, please include the information

Response to Comment:

We added a reference paper about the primer as [19].

12. COMMENT:

L152: is the “table.qzv” include in any supplementary data? If not, delete this part

Response to Comment:

We deleted the “table.qzv” as you advised.

13. COMMENT:

L152: What was the sequence similarity threshold for OTUS? 97% Please include. For the future, please revise ASVs approach, that is more convenient for collecting the heterogeneity of a microbiome

Response to Comment:

The sequence similarity threshold for OTUS was 99 %.

ASVs was extracted using DaDa2, and total error rate was calculated by pooling all sample.

We added it as you advised. Please refer to line 161.

14. COMMENT:

L157: please correct “All P values were corrected…”

Response to Comment:

We corrected it as you advised. Please refer to line 167.

15. COMMENT:

M&M: Please include for all the software’s and pipelines used their respective cites.

Response to Comment:

In this paper, we have already included all the software’s and pipelines used their respective cites in the M&M. Please refer to line 146-148, 152-153, 163-168, 171-172, 174-175, 178-179, 185, 192-194, 202, 211.

16. COMMENT:

L179: did the authors mean “raw”? if yes, please correct

Response to Comment:

We corrected it as you advised. Please refer to line 189.

17. COMMENT:

L182: Please correct “were aligned”

Response to Comment:

We corrected it as you advised. Please refer to line 192-193.

18. COMMENT:

L187: this is not for microbial, this is for bacterial…

Response to Comment:

We corrected it as you advised. Please refer to line 197.

19. COMMENT:

L208: Please correct “the animal protocols were approved…”, or The animals protocol was approved, deeding on what the authors do want to explain

Response to Comment:

We corrected it as you advised. Please refer to line 219.

20. COMMENT:

L215-218: the initial text is more related with discussion or introduction. Results must be short and concise.

Response to Comment:

The initial text provides information for setting the direction of the experiment afterward, and if it is removed, it will not be possible to explain why the experiment was conducted in this way, so I think it is better to leave it as it is.

21. COMMENT:

L253-260: This must be included in material and methods section, not in results. Please change

Response to Comment:

Except for the overlapping content, the remaining parts have been integrated into 2.5.

22. COMMENT:

L307: please correct “the mice clustered according to the weight gain groups”

Response to Comment:

We corrected it as you advised. Please refer to line 313.

23. COMMENT:

L367: please specify why is obvious

Response to Comment:

We erased “ obvious” in the sentence.

24. COMMENT:

Please do not consider the term “up or down regulated” this is not transcriptomics.

Response to Comment:

We corrected the term "up-regulated" to "abundant" in the manuscript and Table S10.

Please refer to line 377 and 380.

Round 2

Reviewer 2 Report

The article has been greatly improved, although some of initial doubts are still present. I'm still not thrilled with the conclusion "This suggests that the regulation of body weight gain by gut microbiome occurs by individual bacteria rather than as a group-wide phenomenon such as Firmicutes or Bacteroidetes in contrary to the previous reports" since it extrapolates results in an organism on OTUS (or ASVs, not species) and does not take into account that the work has not been done at the level of metagenomics or strain analysis, which could provide data in this regard. Authors would need to reduce the weight of this conclusion, including "may occur" or "might occur" instead of "occurs", for example

Author Response

Reviewer 2

Major concerns

The article has been greatly improved, although some of initial doubts are still present. I'm still not thrilled with the conclusion "This suggests that the regulation of body weight gain by gut microbiome occurs by individual bacteria rather than as a group-wide phenomenon such as Firmicutes or Bacteroidetes in contrary to the previous reports" since it extrapolates results in an organism on OTUS (or ASVs, not species) and does not take into account that the work has not been done at the level of metagenomics or strain analysis, which could provide data in this regard. Authors would need to reduce the weight of this conclusion, including "may occur" or "might occur" instead of "occurs", for example

Response to Comment:

We agree with your comment. Thank you very much for your precise comment. We corrected it as your advised. Please refer to line 361. Thank you.